# Evaluation of New Cardiac Damage Biomarkers in Polytrauma: GDF-15, HFABP and uPAR for Predicting Patient Outcomes

**DOI:** 10.3390/jcm13040961

**Published:** 2024-02-08

**Authors:** Aileen Ritter, Lorenz Lötterle, Jiaoyan Han, Miriam Kalbitz, Dirk Henrich, Ingo Marzi, Liudmila Leppik, Birte Weber

**Affiliations:** 1Department of Trauma-, Hand- and Reconstructive Surgery, University Hospital Frankfurt, Goethe-University, 60596 Frankfurt am Main, Germany; l.loetterle@stud.uni-frankfurt.de (L.L.); s9745753@stud.uni-frankfurt.de (J.H.); d.henrich@trauma.uni-frankfurt.de (D.H.); marzi@trauma.uni-frankfurt.de (I.M.); lleppik@yahoo.com (L.L.); bi.weber@med.uni-frankfurt.de (B.W.); 2Department of Trauma and Orthopedic Surgery, University Hospital Erlangen, Friedrich-Alexander-University Erlangen-Nuremberg, 91054 Erlangen, Germany; m.kalbitz@ews-mmc.de

**Keywords:** markers of cardiac damage, extracellular vesicles, troponin, exosomes, miRNA, HFABP, GDF-15, uPAR, miR-21

## Abstract

**Background:** Polytrauma is one of the leading mortality factors in younger patients, and in particular, the presence of cardiac damage correlates with a poor prognosis. Currently, troponin T is the gold standard, although troponin is limited as a biomarker. Therefore, there is a need for new biomarkers of cardiac damage early after trauma. **Methods:** Polytraumatized patients (ISS ≥ 16) were divided into two groups: those with cardiac damage (troponin T > 50 pg/mL, n = 37) and those without cardiac damage (troponin T < 12 pg/mL, n = 32) on admission to the hospital. Patients’ plasma was collected in the emergency room 24 h after trauma, and plasma from healthy volunteers (n = 10) was sampled. The plasma was analyzed for the expression of HFABP, GDF-15 and uPAR proteins, as well as miR-21, miR-29, miR-34, miR-122, miR-125b, miR-133, miR-194, miR-204, and miR-155. Results were correlated with patients’ outcomes. **Results:** HFABP, uPAR, and GDF-15 were increased in polytraumatized patients with cardiac damage (*p* < 0.001) with a need for catecholamines. HFABP was increased in non-survivors. Analysis of systemic miRNA concentrations showed a significant increase in miR-133 (*p* < 0.01) and miR-21 (*p* < 0.05) in patients with cardiac damage. **Conclusion:** All tested plasma proteins, miR-133, and miR-21 were found to reflect the cardiac damage in polytrauma patients. GDF-15 and HFABP were shown to strongly correlate with patients’ outcomes.

## 1. Introduction

Trauma is one of the leading mortality factors in younger patients [1]. In particular, polytrauma is associated with high mortality, as reflected in the definition of the term “polytrauma” as injuries or the combination of injuries that are life-threatening [2]. Approximately 45% of polytrauma patients suffer from thoracic trauma. In addition to acute lung injury, cardiac damage correlates with poor outcomes and prolonged hospitalization after chest trauma [3,4].

Polytraumatized patients need to be diagnosed as fast as possible to recognize the injury pattern, prevent adverse events, and make therapeutic decisions. Therefore, diagnostics in the emergency room are based on clinical examination, laboratory markers, and radiological imaging methods such as X-rays and CT scans. In general, troponin T is the most commonly used parameter and is considered as the gold standard for cardiac damage in combination with an ECG (according to S3 guidelines’ definition of polytrauma). In trauma patients, higher troponin levels correlated with injury severity score (ISS) and increased mortality [5]. Higher troponin levels in the intensive care unit (ICU) have been associated with non-survival and the need for catecholamines [5]. Troponin was elevated as early as 3 h after traumatic cardiac damage in a porcine polytrauma model. This elevation was associated with functional depression of the heart measured by echocardiography [6]. At first sight, troponin seemed to be a reliable prognostic marker in polytraumatized patients to diagnose cardiac damage.

However, there are limitations of troponin as a biomarker, such as the reliance on its expression on kidney function. Friden et al. (2017) showed that troponin levels are two-fold elevated in patients with 50% loss of renal function [7]. Additionally, a single troponin measurement often lacks significance and should be interpreted in the context and with regard to the overall troponin dynamics. A second measurement is often needed if the first troponin measurement falls within the gray area (12–50 pg/mL) of the detection method [5]. Caused by the limitations of troponin, there is a need for alternative markers to detect cardiac damage after trauma and predict patients’ outcomes.

Several new potential protein and miRNA cardiac damage biomarkers are described in recent studies focusing on myocardial infarction or heart failure. Thinking that such markers could be also used to detect cardiac damage in polytrauma patients, we selected proteins and miRNAs that had previously been associated with cardiac damage but that had not been assessed in polytraumatized individuals before.

One potential biomarker of cardiac injury could be heart-type fatty acid binding protein (HFABP). In the blood, this protein peaks earlier than troponin T and could therefore be an earlier marker of cardiac dysfunction. The serum level of HFABP begins to rise within 1 h after cardiac injury and peaks after 4–6 h [8]. With HFABP, the clinicians might detect cardiac damage earlier in admission to the emergency department and without the need for process control. Nevertheless, it is less specific because of its simultaneous expression in the skeletal muscles [9]. Previously, we demonstrated a systemic increase in HFABP after experimental polytrauma, which was associated with a decrease in ejection fraction and a shortened fraction of the left ventricle in pigs [6].

Other promising candidates for diagnosis of cardiac damage are growth/differentiation factor 15 (GDF-15) and urokinase plasminogen activator surface receptor (uPAR). Andersson et al. showed that higher levels of GDF-15 and uPAR are associated with sudden cardiac arrest [10]. In infarcted mice, GDF-15 expression was increased in the heart within hours after myocardial infarction, and its serum concentration was shown to be highly associated with an unfavorable outcome [11]. Next to GDF-15, the soluble form of uPAR has been identified as an independent predictor of all-cause death in patients with coronary artery disease [12]. Caused by the connection between uPAR or GDF-15 and the ischemic cell death of cardiomyocytes, a relationship with trauma-induced cell death might also be possible. However, the association of both proteins with traumatic cardiac damage has not been studied yet.

In addition, miRNAs have been recognized as potential biomarkers of heart failure. For example, Zhang et al. published data showing an association of circulating miR-21 with heart failure [13]. In patients with coronary artery disease, the expression of miR-21 in plasma samples progressively increased in those with single-, dual- and multivessel occlusion. This suggests the potential diagnostic capability of miRNAs for cardiac damage [14]. Furthermore, multiple studies have focused on the identification of biomarker miRNAs in small extracellular vesicles (EVs) or exosomes in recent years. EVs are small phospholipid-layered vesicles that were brought to the attention of researchers back in 1985 [15]. EVs are released under physiological and pathological conditions, which makes them of broad interest as cellular environment markers. Exosomes are the smallest group of EVs, with diameters ranging from 50 to 300 nm, known to play a role in intercellular communication [16]. Exosomes have been identified in various body fluids such as plasma, liquor, urine, lung fluid, pleural effusion, and saliva, making them an easily accessible source of biomarkers [17,18,19,20]. It has been demonstrated that miRNAs inside exosomes are protected from degradation and can influence the synthesis of proteins in the target cells [16].

The present study aimed to evaluate HFABP, GDF-15, and uPAR as well as systemic and exosomal miRNAs as potential markers of trauma-induced cardiac damage in critically ill patients.

## 2. Materials and Methods

**Study design:** The present study included 69 polytraumatized patients with ISS ≥ 16 and systemic troponin T concentrations categorized as either high (>50 pg/mL, high group n = 37) or low (<12 pg/mL, low group, n = 32), as measured by a highly sensitive electrochemiluminescence immunoassay (ECLIA, Roche, Rotkreuz, Switzerland). Blood samples were collected immediately at the emergency room (ER) and 24 h after trauma and kept on ice; plasma was obtained by centrifugation at 3500× *g*, 15 min 4 °C [21]. The control group included plasma samples from 10 healthy volunteers (no medication, no pre-existing illnesses, age > 18 years). These samples were treated in the same way as the samples of the polytraumatized patients.

**Ethical approval** was obtained from the Local Ethics Committee of the University of Frankfurt (approval ID 89/19). Written informed consent was collected from the patients or their legal guardians. Clinical outcome parameters were collected from the digital case file, and these parameters were correlated with the potential new biomarkers.

**ELISAs:** Systemic concentrations of HFABP, GDF-15, and uPAR were measured in plasma samples using ELISA Kits (DY1678, DGD150, DUP00; R&D systems, Wiesbaden, Germany). Measurements were performed according to manufacturer’s instructions.

**miRNA expression analysis:** miRNA was isolated using miRNeasy serum/plasma kit (Qiagen Inc., Hilden, Germany) from 100 µL of plasma or 100 µL of exosomes miRCURY LNA RT Kit (Qiagen Inc., Hilden, Germany) used for cDNA synthesis. RT-qPCR amplification was carried out with the miRCURY SYBR Green PCR Kit (Qiagen Inc., Hilden, Germany) and CFX96 Real-Time PCR Detection System (BioRad, Puchheim, Germany). Cycling conditions were as follows: 1 cycle with 3 min at 95 °C, and 40 cycles with 10 s at 95 °C and 50 s at 56 °C. All reactions were performed according to manufacturer’s instructions. The sample volume for miRNA analysis was standardized. The following commercially available primers were used: cel-miR-39-3p, hsa-miR-204-5p, has-miR-214-5p, has-miR-155-5p, has-miR-34a-5p, has-miR-194-5p, has-miR-125b-5p, and has-miR-29a-3p (miRCURY LNATM miRNA PCR Assay, Qiagen Inc., Hilden, Germany). For relative quantification of target miRNA, the delta Ct method (2−∆Ct) was used and normalization was conducted by using the Ct values of spike-in cel-miR-39-3p.

**Exosome isolation and analysis:** Exosome isolation was performed with 100 µL of plasma using an Exo-Spin^TM^ Kit (Cell guidance systems, Cambridge, UK). Before isolation, plasma was cleared by 30 min of centrifugation at 16,000× *g*, (4 °C). The isolation is based on size exclusion chromatography (SEC), and particle-free PBS was used for elution of the exosomes from the SEC columns.

**Exosomal troponin T ELISA:** For exosomal ELISA, the isolated exosomes were lysed with M-PER^®^ Mammalian Protein Extraction Reagent (Thermo Fisher Scientific, Waltham, MA, USA) with 1× Proteinase inhibitor Halt^TM^ (Thermo Fisher Scientific, Waltham, MA, USA) (1:1, 15 min incubation time). The troponin T concentration was measured with a highly sensitive troponin T ELISA Kit (ABIN6960198, antibodies-online.com, Aachen, Germany), according to the manufacturer’s instructions.

**Statistical analysis:** GraphPad Prism 9 (Dotmatic, San Diego, CA, USA) was used for all statistical analysis. Data were analyzed via a Kruskal–Wallis test followed by Dunn‘s multiple comparisons test. For the statistical analysis of two groups, the Mann–Whitney test was performed. Correlation analysis was carried out with the Spearman rank correlation. Results with *p* ≤ 0.05 were considered significant. Data are presented as mean ± standard error of the mean (SEM).

## 3. Results

First, the plasma levels of potential new protein biomarkers (HFABP, GDF-15 and uPAR) in the healthy control group and in polytraumatized patients at the ER and 24 h after trauma were measured. Plasma concentrations of all three proteins at both timepoints were significantly higher in the subgroup of polytraumatized patients with high troponin, as compared to healthy controls. However, while HFABP protein expression decreased with time, the expression of both uPAR and GDF-15 proteins was higher at the 24 h time point than at the ER (Figure 1).

The level of plasma troponin T was correlated with HFABP, uPAR, and GDF-15 plasma levels in all analyzed patients, revealing a positive correlation between troponin T and all three parameters (r = 0.59, *p* < 0.0001; r = 0.6, *p* < 0.0001 and r = 0.6, *p* < 0.0001; Figure 2).

In addition, we performed a correlation analysis of HFABP, uPAR, and GDF-15 concentrations with three clinical outcome parameters: the time on ICU/IMC, overall time in the hospital, and ventilation time (Figure 3). Plasma levels of HFABP (r = 0.44), uPAR (r = 0.42) and GDF-15 (r = 0.43) were moderately correlated with time on the ICU/IMC. A mild correlation was observed between all three biomarkers (r = 0.36, r = 0.29 and r = 0.33) and the overall time in hospital. At the same time, HFABP was found to correlate mildly with the ventilation time (r = 0.32), whereas uPAR and GDF-15 clearly showed moderate correlation with these clinical parameters (r = 0.56 and r = 0.66) (Figure 3).

The correlation analysis of biomarker concentrations, the need for catecholamines, and survival showed that higher systemic levels of HFABP, GDF-15, and uPAR were associated with the need for catecholamine therapy. Regarding survival, HFABP was the only biomarker which was significantly increased in non-survivors (Figure 4).

Among the various miRNAs discussed in the literature as potential biomarkers of cardiac damage, we selected the following for expression analysis: miR-204-5p, miR-214-5p, miR-155-5p, miR-34a-5p, miR-194-5p, miR-125b-5p, miR-29a-3p. Overall increased expression of most of the selected miRNAs was observed upon admission to the ER in polytraumatized patients (Figure 5). In detail, miR-21 (*p* < 0.05) and miR-133 (*p* < 0.01) exhibited a significant elevation in the high-troponin sub-group of polytrauma patients as compared to healthy controls, while no significant difference was observed between the low-troponin sub-group and healthy controls. Expression of miR-29, miR-34, miR-125b, and miR-204 was elevated upon ER admission in both the low- and high-troponin sub-groups of patients. Expression of miR-194 and miR-155 measured in the high-troponin sub-group of patients at two different time points differed significantly; however, there was no difference among the levels in patients and controls (Figure 5).

In addition, we compared exosomal levels of selected miRNAs among the groups (Figure 6). In contrast to systemic expression, no significant difference of exosomal miRNA concentrations were found between polytrauma patients and healthy controls. None of the miRNAs showed any association with cardiac damage in polytraumatized patients. miR122 showed a significant difference between time points, but no difference in comparison with the control.

We analyzed the level of troponin T inside the exosomes in our groups of patients and compared it to controls (Figure 7). No significant difference in exosomal troponin concentrations measured at both time points was observed between the high- and low-troponin groups of polytraumatized patients, as well as the healthy controls (Figure 7A). Moreover, we found a negative correlation between exosomal and systemic troponin concentrations in our patients (r = −0.51, *p** = 0.02, Figure 7B), as well as a negative correlation between exosomal troponin concentration and ISS (r = −0.4, *p* = 0.06, Figure 7C). No difference in exosomal troponin concentration was observed between patients with and without the need for catecholamines (Figure 7D). Finally, we analyzed the correlation between exosomal troponin and outcome parameters. No strong correlation between troponin T in exosomes and ventilation time (Figure 7E), time in the hospital (Figure 7F), and time on the ICU/IMC (Figure 7G) was observed.

## 4. Discussion

The aim of the present study was to evaluate new protein and miRNA markers of cardiac damage in polytrauma patients. In this study, 69 polytrauma patients with ISS ≥ 16 were divided into two sub-groups according to their initial troponin T level (high- and low-troponin sub-groups), and potential biomarker expression in blood samples was measured and compared with healthy controls.

Our results demonstrated a significant increase in all three analyzed proteins (HFABP, GDF-15 and uPAR) in the high-troponin subgroup of polytrauma patients. This indicates an association of these new biomarkers with cardiac damage. HFABP, a member of the lipid-binding proteins superfamily, plays a crucial role in the intracellular transport of fatty acids to the sites of metabolic conversion [8]. One of the important advantages of HFABP as a cardiac damage biomarker is that it can be detected earlier as troponin T. HFABP is released from damaged cells into the blood very early. The increase in the blood starts 2–3 h after the onset of chest pain in cases of myocardial infarction. The values are demonstrated to return to baseline values after 12–24 h [22,23]. The concentration of H-FABP is significantly influenced by renal clearance and thus has limitations in its usefulness for patients with renal dysfunction. Serum HFABP and myoglobin levels were significantly elevated in hemodialysis patients and reduced by 30–40% during hemodialysis [24]. In the context of cardiac damage after trauma, HFABP was described in different animal studies as the earliest marker of cardiomyocyte sinking. In an experimental porcine multiple trauma model, an increase in systemic HFABP concentration was associated with depression of cardiac function, including reduction in ejection and shortening fractions [6]. HFABP, troponin, and extracellular histones were found to be elevated early after multiple traumata and returned to baseline after 24 h. In a murine model of traumatic hemorrhagic shock with a resuscitation phase, a persistent and progressive loss of cardiac function was associated with increasing myocardial injury and elevation of HFABP [25]. Furthermore, in an in vitro study with murine HL-1 cells, it was shown that treatment with a polytrauma cytokine cocktail induced a significant elevation of HFABP and troponin in the cell culture supernatant [26]. In a previous study, HFABP was suggested as a prognostic marker in trauma patients due to its significant specificity and sensitivity in predicting complete recovery in patients with mild TBI [27]. De’Ath et al. described elevated levels of HFABP at 0, 24 and 72 h after trauma and found this elevation to be associated with the appearance of adverse cardiac events as one example of trauma-induced secondary cardiac injury. HFABP levels were 3.2 times higher in patients with adverse cardiac effects than in patients without [28]. Nevertheless, HFABP expression is not limited to cardiac tissue, and this protein can be found in the skeletal muscles, kidneys, mammary glands, testes, lungs and stomach, which limits its potential use as a cardiac damage biomarker in polytrauma patients [29]. Based on our findings, which include a strong correlation between HFABP and troponin T levels, a moderate correlation of HFABP with the time on the ICU/IMC, and its ability to differentiate between survivors and non-survivors, this protein exhibits strong potential as a marker for cardiac injury. Future studies should verify association of HFABP with cardiac functions detected by echocardiography and/or Cardio-MRI.

Among all the markers evaluated in the present study, GDF-15 proved to be the most effective in predicting outcome parameters in polytraumatized patients. It exhibited a moderate correlation with the time on the ICU/IMC and in-hospital time, and a strong correlation with ventilation time and the need for catecholamines. GDF15 is not only expressed in the heart, but also at different levels in various human tissues including the placenta, kidneys, lungs, pancreas, skeletal muscles, liver, and brain [30]. Therefore, GDF-15 is not heart-specific and is, e.g., associated with chronic kidney diseases. In particular, with regard to the heart, GDF-15 predicts morbidity and mortality in stable coronary heart disease [31]. A cut off-value of GDF-15 was described at 1808 ng/mL in patients with a non-St-elevation acute coronary heart syndrome [32]. Interestingly, after a month of an acute chest pain, GDF-15 was also described as a prognostic marker for non-ST elevation acute coronary syndrome [33]. GDF-15 is a cytokine belonging to the transforming growth factor (TGF-) family, known to be highly up-regulated in stress and inflammatory conditions [34]. In a previous in vivo study, GDF-15 has been correlated with myocardial injury and pressure cardiac overload. Sequential measurements of GDF-15 in patients with acute heart failure revealed that GDF-15 independently and dynamically predicts the risk of an adverse outcome (all-cause mortality and heart failure rehospitalization) during 1 year of follow-up [35]. Interestingly, GDF-15 is also discussed in the literature as a marker of kidney function and for its protective function for chronic kidney injury prevention [36]. Some studies described GDF-15 in the context of cardiotoxicity in cancer treatment with doxorubicin, taxanes or trastuzumab [37] or due to radiation [38]. Circulating GDF-15 was associated with subclinical brain injury and cognitive impairment in patients, but not with the appearance of stroke [39]. Overall, GDF-15 has been previously described as a marker of cardiac and vascular stress, and our findings emphasize its importance in polytrauma-patients.

suPAR or soluble urokinase plasminogen activator receptor has been described as an inflammation and immune activation -biomarker [40]. It correlates positively with several inflammatory biomarkers, including CRP, IL-6 and TNFα [41], and has been investigated as a marker of kidney- and heart- injury, as well as sepsis. suPAR or soluble urokinase plasminogen activator receptor has been described as an inflammation and immune activation -biomarker [33]. It correlates positively with several inflammatory biomarkers, including CRP, IL-6 and TNFα [34], and has been investigated as a marker of kidney- and heart- injury, as well as sepsis. Therefore, uPAR is also not heart-specific. Additionally, suPAR seemed not to be a marker of acute changes, because suPAR levels were described to be unaltered for the first 24 h after myocardial infarction [42]. Furthermore, uPAR measured shortly after trauma was not associated with the severity of the trauma; however, later, it was higher in non-survivors compared with those who survived [43]. In patients with chronic heart failure, suPAR was shown to be independently associated with 96-month mortality [44]. suPAR was found to differentiate between heart failure and non-heart failure patients and to predict patients’ outcome and risks [45]. In type I diabetes patients with normal left ventricular ejection fraction and without known heart diseases, suPAR was associated with early systolic and diastolic myocardial impairment, which suggests it could be used as a predictor of cardiac complications in these patients [46]. No previous study has investigated the correlation between expression of GDF-15 and cardiac injury in patients with chest or polytrauma. Our findings indicate that plasma concentrations of uPAR strongly correlate with troponin T concentrations in polytraumatized patients. Plasma concentrations of uPAR also moderately correlate with outcome parameters such as time on the ICU/IMC and ventilation time, suggesting a clear association with cardiac damage in polytrauma.

Among all analyzed miRNAs, only miR-21 and miR-133 were found to be differentially expressed in polytraumatized patients with cardiac damage, whereas all other miRNAs were elevated upon admission to ER in all patients. These results are in accordance with previous studies mentioning the higher serum level of miR-21 in patients with heart failure [13]. In a rat model of drug-induced cardiac injury, miR-21 was specifically localized to inflammatory cell infiltrates in the heart and therefore was suggested as a biomarker of cardiotoxicity [47]. In patients with coronary heart disease, expression of plasma miR-21 was found to be progressively higher in single-, dual-, and multivessel occluded coronary artery disease patients [14]. In heart failure patients, the level of plasma miR-21 was higher than in healthy controls, but no association between this miRNA and the prognostic outcome parameters was described [48]. In a murine model of acute cardiac infarction, suppression of miR-21 was shown to decrease the infarct size, increase ejection fraction, and induce fractional shortening [49]. It has been previously described that miR-21 plays an important role in many diseases such as bone fractures, traumatic brain injury and spinal cord injury. In spinal cord injury, miR-21 derived from exosomes was shown to inhibit the expression of PTEN/PDCD4 and suppress neuron cell death [50]. Furthermore, in mice with cardiac infarction, transfection of miR-21 agomir led to decreased apoptosis mediated via the miR-21-PTEN/AKT-p-p38-caspase-3 pathway [49]. Taken together, miR-21 seems to play an important role in cardiac damage, which is not fully understood. The present analysis for the first time described miR-21 in polytraumatized patients with significant troponin T increase at the ER. Further experiments are needed to provide detailed information about miR-21 signaling pathways in polytrauma patients.

Another miRNA whose expression was associated with an increase in troponin T at the ER and 24 h after trauma was miR-133. In accordance with our findings, a significant increase in this miRNA was shown in the serum of patients with cardiac ischemia [51]. Conversely, in the tissues of infarcted hearts, miR-133 was found to be significantly downregulated [52]. A simultaneous increase in systemicmiR-133 expression and a decrease in local (heart tissue) miR-133 expression suggest that as a result of myocardial ischemia, these miRNAs might be released from ischemic cardiomyocytes into the blood. The important role of this miRNA in maintaining and regaining cardiac functionality has been suggested in the literature. For instance, miR-133 has been shown to restrict injury-induced cardiomyocyte proliferation in zebrafish [53] and improve cardiac contractile function in a congestive heart failure rabbit model [54]. Our data suggest the involvement of this miRNAs in cardiac damage processes in polytrauma patients, although the exact role and the mechanism should be investigated.

For all other analyzed miRNAs, we were not able to show any correlation with cardiac damage in polytrauma patients. Although miR-125b was previously associated with cardiac diseases like heart failure [55] or myocardial damage [56], our results suggest that these miRNAs are rather polytrauma-specific and not cardiac damage-specific. Similarly to our findings, upregulation of miR-125b and miR-204 was previously described in patients with a traumatic hemorrhagic shock and was associated with endoplasmic reticulum stress [57]. The role of miR-29 in the context of cardiac remodeling was described in chronic cardia stress patients [58], but not in polytrauma patients. miR-34 was shown to decrease cardiomyocytes HL-1 cells’ viability and promote production of proinflammatory cytokines in a doxirubicin-induced myocardial cell damage model [59], but less is known about its function in polytrauma patients. In summary, our finding suggests that miR-21 and miR-133 could be the candidate biomarkers of cardiac damage in polytrauma patients, while miR-125b, -204, -29, and -34 should be further investigated as potential diagnostic markers of polytrauma.

In recent years, exosomes have been discussed as an alternative source of biomarker proteins and miRNAs molecules [60] preserved from systemic degradation. We analyzed the levels of selected miRNAs and troponin T in exosomal cargo in our patients and control groups. We found that exosomal expression of selected miRNAs differs significantly from the systemic expression and does not correlate with cardiac damage. Interestingly, exosomal troponin T negatively correlates with systemic troponin T concentration, but does not show any correlation with any of the clinical parameters, and does not differ between polytrauma patients and healthy controls. Therefore, we could speculate that after trauma, most troponin is released directly into the blood, while only a minor amount of it is secreted via exosomes. In contrast to our findings, a previous study demonstrated a significant increase in exosomal troponin T in patients with myocardial infarction as compared to those without [61]. In addition, in patients with coronary artery bypass graft, the systemic troponin concentration was found to correlate with the total amount of exosomes measured by a nanoparticle tracking assay (NTA) [62]. It was also suggested that troponin could be released from damaged cardiomyocytes in apoptotic bodies [63]. In our study, the fraction of apoptotic bodies within EVs was excluded from the evaluation, which could partially explain the discrepancy in the results. The difference in patient cohorts among the study and differences in mechanisms of injury could be another reason for this. It is possible that cardiomyocytes in polytrauma patients are “less” damaged or damaged “differently” than in infarct patients, influencing the appearance of troponin in exosomal cargo. The present analysis showed the potential for novel protein and miRNA biomarkers to detect cardiac damage in polytraumatized patients. Future studies which systematically analyse cardiac function in polytraumatized patients by measuring echocardiography are required. The results of these studies should be further correlated with the biomarkers discussed here, and in cases of good correlation, HFABP, GDF-15, or uPAR, as well as miR-133 or miR-21, could be considered for implementation in standard laboratory measurements carried out in the emergency room. According to S3 polytrauma guidelines in Germany, troponin T and ECG measurements are currently used to detect cardiac contusion in trauma patients. One of our new cardiac markers could be a good addition to troponin measurements and might help to detect cardiac damage early in the emergency room. A combination of troponin and one of the new biomarkers could help to overcome some of troponin’s limitations. Moreover, novel markers should be included in routine follow-up laboratory diagnostics on the ICU to facilitate the identification of secondary cardiac damage resulting from inflammation or sepsis throughout the in-hospital stay.

## 5. Conclusions

HFABP, uPAR, and GDF-15 were increased in polytraumatized patients with cardiac damage detected by increased troponin T levels. GDF-15 and HFABP were shown to correlate with parameters of patients’ outcomes. Analysis of systemic miRNA concentrations showed a significant increase in miR-133 (*p* < 0.01) and miR-21 (*p* < 0.05) in patients with cardiac damage. Other miRNAs were shown to be significantly elevated in all polytrauma patients at the ER time point. In contrast to systemic expression, no significant differences in exosomal miRNA concentrations were found between polytrauma patients and healthy controls, and none of the miRNAs showed any association with cardiac damage. Further analysis will be needed to verify if GDF-15 and HFABP are useful tools for detecting functional impairment of the heart in polytraumatized patients. Analysis of echocardiographic measurements and correlation with the present biomarkers are needed.

## Figures and Tables

**Figure 1 jcm-13-00961-f001:**
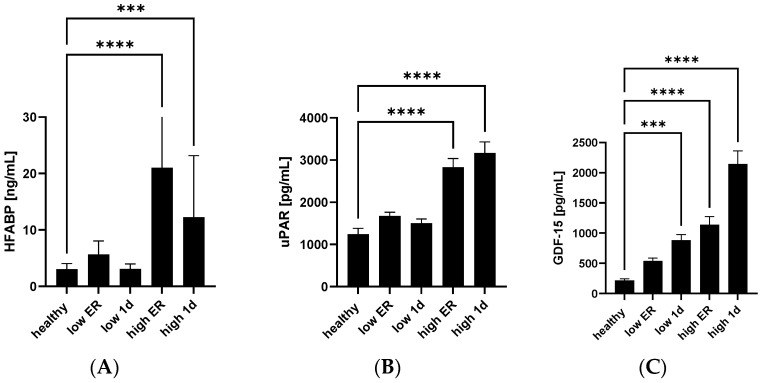
HFABP, uPAR, and GDF-15 were increased in polytraumatized patients with elevated troponin T concentrations. Systemic HFABP (heart-type fatty acid binding protein) (**A**), uPAR (urokinase plasminogen activator surface receptor) (**B**) and GDF-15 (growth/differentiation factor 15) (**C**) were measured in plasma of polytraumatized patients with low and high troponin T. Measurements were carried out at the ER (emergency room) and 24 h after trauma. **** *p* < 0.0001; *** *p* < 0.001, high group, n = 37, low group, n = 32, healthy group, n = 10.

**Figure 2 jcm-13-00961-f002:**
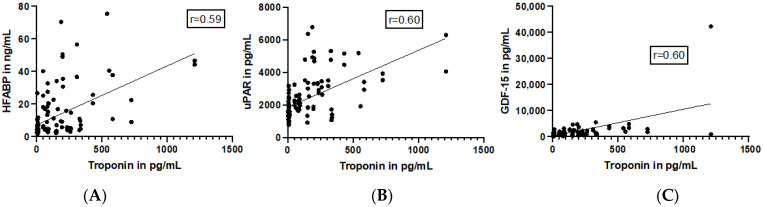
Positive correlation of Troponin T and HFABP, uPAR and GDF-15 in polytraumatized patients. A positive correlation was found among the plasma levels of troponin T and HFABP (heart-type fatty acid binding protein (**A**), uPAR (urokinase plasminogen activator surface receptor) (**B**), and GDF-15 (growth/differentiation factor 15) (**C**) in all polytraumatized patients (n healthy = 10, n low = 12, n high = 37).

**Figure 3 jcm-13-00961-f003:**
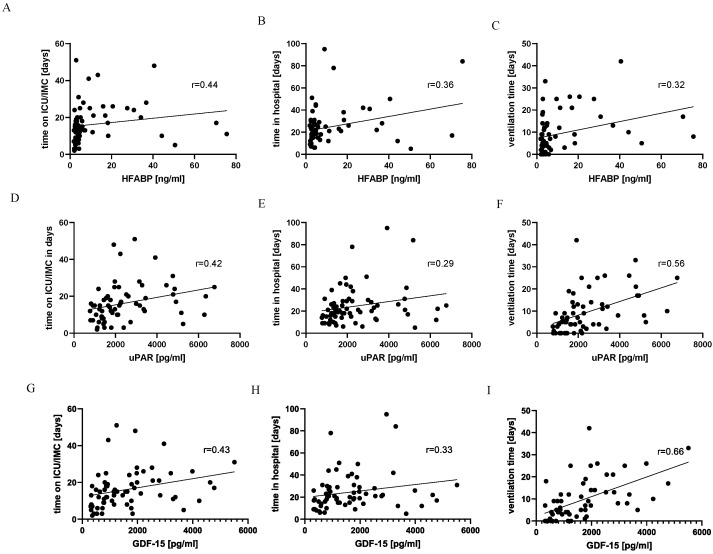
HFABP, uPAR, and GDF-15 showed correlation with clinical parameters. Correlation analysis of HFABP (heart-type fatty acid binding protein) and various clinical outcome parameters: (**A**) time on ICU/IMC, (**B**) time in hospital, and (**C**) ventilation time. The same correlation analysis was performed for uPAR (urokinase plasminogen activator surface receptor) (**D**–**F**). GDF-15 (growth/differentiation factor 15) was also correlated with (**G**) time on ICU/IMC; (**H**) time in hospital; and (**I**) ventilation time. n healthy = 10, n low = 12, n high = 37.

**Figure 4 jcm-13-00961-f004:**
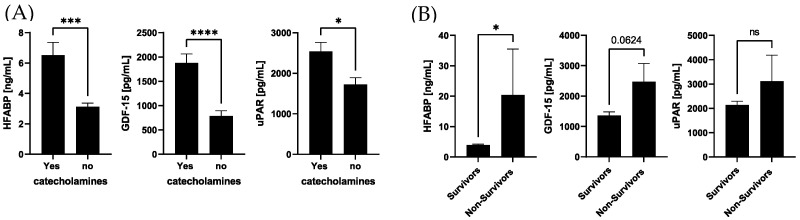
Correlation analysis among biomarker concentrations and the need for catecholamines and survival. (**A**) Levels of HFABP, GDF-15, and uPAR were analyzed depending on the need for catecholamine and (**B**) in non-survivors compared to survivors. **** *p* < 0.0001; *** *p* < 0.001, * *p* < 0.05, ns = not significant, healthy group, n = 10, low group, n = 12, high group, n = 37.

**Figure 5 jcm-13-00961-f005:**
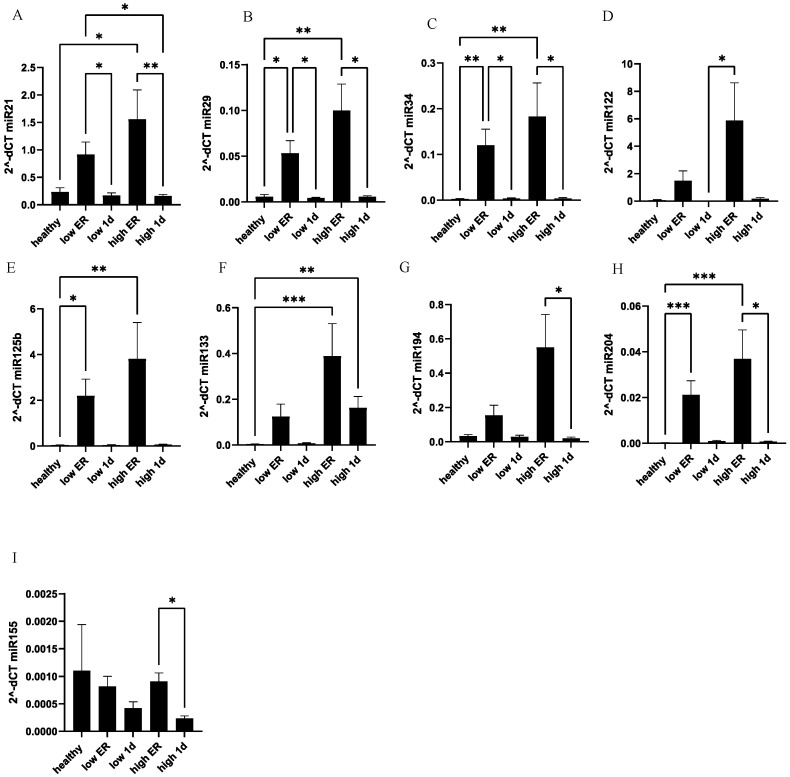
MiR-21 and miR-133 show significant elevation in polytrauma patients with high troponin concentrations as compared to healthy controls. At the ER and after 24 h, systemic expression of miR-21 (**A**), miR-29 (**B**), miR-34 (**C**), miR-122 (**D**), miR-125b (**E**), miR-133 (**F**), miR-194 (**G**), miR-204 (**H**), and miR-155 (**I**) were measured in polytraumatized patients in both the high- and low-troponin groups. *** *p* < 0.001; ** *p* <0.01; * *p* < 0.05.

**Figure 6 jcm-13-00961-f006:**
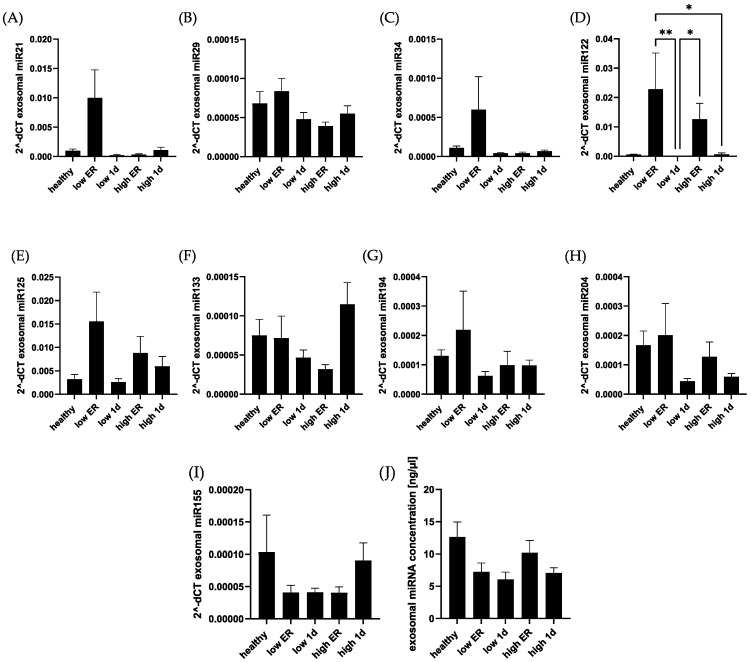
Exosomal miRNA concentration did not differ significantly among polytraumatized patients and healthy controls. Exosomal concentrations of miR-21 (**A**), miR-29 (**B**), miR-34 (**C**), miR-122 (**D**), miR-125b (**E**), miR-133 (**F**), miR-194 (**G**), miR-204 (**H**), and miR-155 (**I**) were compared among healthy controls and polytrauma patients. Exosomal miRNA concentration in total was compared in the mentioned groups (**J**). ** *p* < 0.01; * *p* < 0.05.

**Figure 7 jcm-13-00961-f007:**
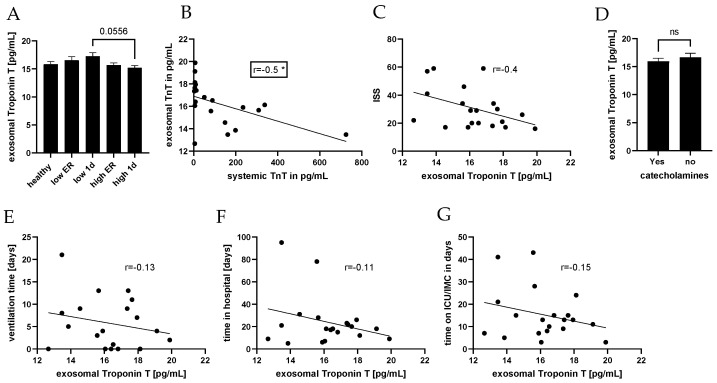
Correlation analysis of exosomal troponin concentrations and clinical parameters. Exosomal levels of troponin were comparable in healthy controls and polytrauma patients (**A**). Exosomal troponin was correlated with the systemic Troponin T concentration (**B**), as well as the injury severity score (iSs) (**C**). Exosomal troponin T correlation was further analyzed in polytraumatized patients with and without the need for catecholamine (**D**). A correlation analysis of exosomal troponin concentration and ventilation time (**E**), time in hospital (**F**), and time on ICU/IMC (**G**) was conducted. Healthy group, n = 10, low, n = 12, high, n = 37, ns = not significant. * *p* < 0.05.

## Data Availability

Data are contained within the article.

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
