# Peer review of "Evaluation of New Cardiac Damage Biomarkers in Polytrauma: GDF-15, HFABP and uPAR for Predicting Patient Outcomes"

_jcm, 2024, doi:10.3390/jcm13040961_

Round 1

Reviewer 1 Report

Comments and Suggestions for Authors

This manuscript exhibits several noteworthy shortcomings:

Introduction and Background:

Original Comment: It lacks a clear introduction and background introduction, and does not clearly state the purpose and significance of the research.

Revision Suggestion: Clearly articulate the research's purpose and significance in the introduction, incorporating background information and relevant research context.

Data and Statistical Analysis:

Original Comment: The article mentioned some research results but did not provide specific data and statistical analysis, lacking empirical support.

Revision Suggestion: Support research findings with specific data and thorough statistical analysis to enhance empirical validity.

Biomarker Selection and Explanation:

Original Comment: The article mentioned some potential biomarkers but did not explain why these markers were selected and their biological significance.

Revision Suggestion: Elaborate on the rationale behind biomarker selection and elucidate their biological significance, particularly in relation to cardiac injury and potential clinical applications.

Research Methods and Experimental Design:

Original Comment: Lack of detailed description of research methods and experimental design prevents readers from understanding the reliability and reproducibility of the research.

Revision Suggestion: Provide a comprehensive description of research methods and experimental design, encompassing sample collection, processing, and analytical techniques to enhance transparency and reproducibility.

Conclusion Section:

Original Comment: The conclusion section is too simple and does not provide sufficient discussion and explanation of the research results.

Revision Suggestion: Enrich the conclusion with detailed discussion and explanation of research results, offering a more comprehensive summary of the findings.

Literature Comparison:

Original Comment: There is a lack of discussion and comparison of other related studies to connect the results of this study with the existing literature.

Revision Suggestion: Integrate a thorough discussion and comparison of related studies to establish connections between the research findings and the existing literature.

Conclusion Detail:

Original Comment: The conclusion should be more detailed and informative.

Revision Suggestion: Enhance the conclusion's depth and informativeness by providing additional details and insights derived from the research findings.

Comments on the Quality of English Language

Grammar and Spelling:

Original Comment: There are some grammatical and spelling errors in the article and need to be proofread and revised.

Revision Suggestion: Conduct a thorough proofreading to correct grammatical and spelling errors, ensuring the article's accuracy and fluency.

Author Response

Re: Manuscript “Evaluation of new cardiac damage- biomarkers in polytrauma: GDF-15, HFABP and uPAR are predicting the patient’s outcome”

Dear Editor,

First of all, we want to thank you and the reviewers for the critical proofreading and the constructive evaluation of our manuscript. After carefully revising our manuscript to resolve the concerns, we believe that it now meets the high standards necessary to be published in the Journal of Clinical Medicine. The detailed point-to-point response to the reviewers’ comments is outlined below.

Sincerely,

Dr. med. Aileen Ritter

Point-by-Point Response to reviewers’ comments:

Reviewer 1

Comment 1: It lacks a clear introduction and background introduction, and does not clearly state the purpose and significance of the research.

Revision Suggestion: Clearly articulate the research's purpose and significance in the introduction, incorporating background information and relevant research context.

Response: Thank you for the opportunity to clarify this point. We revised the whole introduction section to make this clear.

Comment 2: The article mentioned some research results but did not provide specific data and statistical analysis, lacking empirical support.

Revision Suggestion: Support research findings with specific data and thorough statistical analysis to enhance empirical validity.

Response: Unfortunately, we cannot trace what the reviewer meant. We did not report any results as “data not shown” and provided data of all performed analysis and statistics in the Figures 1-7.

Comment 3: The article mentioned some potential biomarkers but did not explain why these markers were selected and their biological significance.

Revision Suggestion: Elaborate on the rationale behind biomarker selection and elucidate their biological significance, particularly in relation to cardiac injury and potential clinical applications.

Response: We appreciated this helpful comment. The rationale behind the selection of biomarkers is now highlighted in the introduction section lines 62-103.

Comment 4: Lack of detailed description of research methods and experimental design prevents readers from understanding the reliability and reproducibility of the research.

Revision Suggestion: Provide a comprehensive description of research methods and experimental design, encompassing sample collection, processing, and analytical techniques to enhance transparency and reproducibility.

Response: We provided detailed information to the methods part of the manuscript.

Comment 5: The conclusion section is too simple and does not provide sufficient discussion and explanation of the research results.

Revision Suggestion: Enrich the conclusion with detailed discussion and explanation of research results, offering a more comprehensive summary of the findings.

Response: In the instructions for authors of the Journal of molecular medicine it says: the conclusion “section is not mandatory but can be added to the manuscript if the discussion is unusually long or complex”. Therefore, we though the conclusion part should serve this purpose and be short and specific. To provide missing explanation of research results, offering a more comprehensive summary of the findings we added text to the end of discussion part:

“HFABP, uPAR and GDF-15 were increased in polytraumatized patients with cardiac damage detected by increased troponin T level. GDF-15 and HFABP were shown to correlate with patients’ outcome parameters. Analysis of systemic miRNA concentrations showed significant increase of miR-133 (p<0.01) and miR-21 (p<0.05) in patients with cardiac damage. Other miRNAs were shown to be significantly elevated in all polytrauma patients at the ER time point. In contrast to systemic expression, no significant difference of exosomal miRNA concentrations were found between polytrauma patients and healthy controls, and none of the miRNAs showed any association with cardiac damage. Further analysis will be needed to verify if GDF-15 and HFABP are useful tools to detect functional impairment of the heart in polytraumatized patients. Analysis of echocardiographic measurements and correlation with the present biomarkers are needed.”

Comment 6: There is a lack of discussion and comparison of other related studies to connect the results of this study with the existing literature.

Revision Suggestion: Integrate a thorough discussion and comparison of related studies to establish connections between the research findings and the existing literature.

Response: In the discussion part of the manuscript, we compared our results with another 28 existing in vitro, in vivo and clinical studies (references 6-54). To further highlight this, we added additional text to the end of discussion (please see the answer to comment 5). Furthermore, we added 9 additional studies to the references mentioned in the discussion section.

Comment 7: The conclusion should be more detailed and informative.

Revision Suggestion: Enhance the conclusion's depth and informativeness by providing additional details and insights derived from the research findings.

Response: Please see the answer to the comment 5

Comment 8: There are some grammatical and spelling errors in the article and need to be proofread and revised.

Revision Suggestion: Conduct a thorough proofreading to correct grammatical and spelling errors, ensuring the article's accuracy and fluency.

Response: Thank you for this comment. The grammatical and spelling proofreading has been performed.

Reviewer 2 Report

Comments and Suggestions for Authors

This study identifies new biomarkers for early detection of cardiac damage in polytrauma patients, revealing that HFABP, uPAR, GDF-15, miR-133, and miR-21 are significantly elevated in those with cardiac damage. These markers, particularly GDF-15 and HFABP, also strongly correlate with patient outcomes. There are several areas the authors should consider for improvement:

1. The study highlights limitations of troponin T as a sole marker for cardiac damage in polytrauma patients and introduces HFABP, uPAR, and GDF-15 as correlated alternatives. The authors should more clearly show the specific added value these biomarkers provide beyond troponin T (e.g., diagnostic accuracy, early detection, and prognostic capability)

2. For Figure 3, the authors could measure troponin T concentrations in conjunction with the three clinical outcome parameters outlined, serving as a positive control

3. In the discussion section, the authors should elaborate on the potential clinical value of the identified biomarkers. Specifically, a detailed exploration of how these biomarkers could be applied in the clinical detection of cardiac damage in polytrauma patients would be valuable. This discussion should ideally include practical implications, potential integration into current diagnostic protocols, and how these biomarkers might improve patient outcomes or treatment strategies

Round 2

Reviewer 1 Report

Comments and Suggestions for Authors

The author addressed most of the Comments. Overall, the author seems to have made an effort to address the reviewer's comments and suggestions. Their responses indicate a commitment to improving the clarity, validity, and readability of their research article.